# Comparative Transcriptomic Analysis Uncovers Genes Responsible for the DHA Enhancement in the Mutant *Aurantiochytrium* sp.

**DOI:** 10.3390/microorganisms8040529

**Published:** 2020-04-07

**Authors:** Liangxu Liu, Zhangli Hu, Shuangfei Li, Hao Yang, Siting Li, Chuhan Lv, Madiha Zaynab, Christopher H. K. Cheng, Huapu Chen, Xuewei Yang

**Affiliations:** 1Guangdong Technology Research Center for Marine Algal Bioengineering, Guangdong Key Laboratory of Plant Epigenetics, College of Life Sciences and Oceanography, Shenzhen University, Shenzhen 518060, China; liangxuliu@icloud.com (L.L.); huzl@szu.edu.cn (Z.H.); sfli@szu.edu.cn (S.L.); 3718aaa@163.com (H.Y.); siting.li@aut.ac.nz (S.L.); LVCHU0510@163.com (C.L.); 15820761674@163.com (M.Z.); 2Shenzhen Key Laboratory of Marine Biological Resources and Ecology Environment, Shenzhen Key Laboratory of Microbial Genetic Engineering, College of Life Sciences and Oceanography, Shenzhen University, Shenzhen 518055, China; 3Longhua Innovation Institute for Biotechnology, Shenzhen University, Shenzhen 518060, China; 4School of Biomedical Sciences, The Chinese University of Hong Kong, Hong Kong 999077, China; chkcheng@cuhk.edu.hk; 5Guangdong Research Center on Reproductive Control and Breeding Technology of Indigenous Valuable Fish Species, Fisheries College, Guangdong Ocean University, Zhanjiang 524088, China; chpsysu@hotmail.com

**Keywords:** Docosahexaenoic acid (DHA), polyunsaturated fatty acids (PUFAs), mutant strain, *Aurantiochytrium* sp., transcriptome

## Abstract

Docosahexaenoic acid (DHA), a *n*-3 long-chain polyunsaturated fatty acid, is critical for physiological activities of the human body. Marine eukaryote *Aurantiochytrium* sp. is considered a promising source for DHA production. Mutational studies have shown that ultraviolet (UV) irradiation (50 W, 30 s) could be utilized as a breeding strategy for obtaining high-yield DHA-producing *Aurantiochytrium* sp. After UV irradiation (50 W, 30 s), the mutant strain X2 which shows enhanced lipid (1.79-fold, 1417.37 mg/L) and DHA (1.90-fold, 624.93 mg/L) production, was selected from the wild *Aurantiochytrium* sp. Instead of eicosapentaenoic acid (EPA), 9.07% of docosapentaenoic acid (DPA) was observed in the mutant strain X2. The comparative transcriptomic analysis showed that in both wild type and mutant strain, the fatty acid synthesis (FAS) pathway was incomplete with key desaturases, but genes related to the polyketide synthase (PKS) pathway were observed. Results presented that mRNA expression levels of *CoAT, AT, ER, DH,* and *MT* down-regulated in wild type but up-regulated in mutant strain X2, corresponding to the increased intercellular DHA accumulation. These findings indicated that *CoAT*, *AT*, *ER*, *DH*, and *MT* can be exploited for high DHA yields in *Aurantiochytrium*.

## 1. Introduction

Owing to the importance of cell membrane function and numerous cellular processes for maintaining health, long-chain polyunsaturated fatty acids (LC-PUFAs) have attracted increasing attention for human health. LC-PUFAs can be classified into two principal families, namely, omega-3 (*n*-3) and omega-6 (*n*-6) fatty acids (FAs) [1]. The typical *n*-3 LC-PUFAs are docosahexaenoic acid (DHA) and eicosapentaenoic acid (EPA), which can strongly influence monocyte physiology. Previous studies have reported that DHA could potently inhibit platelet aggregation [2], reduce hemoglobin formation [3], treat cardiovascular diseases [4], and prevent osteoporosis [5]. Currently, fatty fish including sardines [6], *Oncorhynchus keta* [7], *Thunnus* [8], etc. is being used as the primary global supply for DHA [9]. However, the industry is severely limited by the original low levels and the instability of *n*-3 LC-PUFAs, which is caused by the fish variation, the climate, and high concentrations of the cholesterol [10]. Eggs naturally contain small amounts of DHA, but new DHA enriched eggs can contain up to 258.2 mg of DHA per egg [11]. Furthermore, marine microalgae including *chrysophyta* [12], *dinoflagellate* [13], and diatom [14], are also regarded as a promising alternative as the primary producer of the EPA and DHA in marine food webs. 

Marine eukaryotes, such as *Thraustochytrids* [15] and *Schizochytrium*, with abundant FA contents, have emerged as promising producers of *n*-3 LC-PUFAs [16]. The FAs content required in the industry is currently 40–45 g/L, and the biomass required is 200 g/L [17]. The fermentation process of the *Schizochytrium* sp. SR 21 was optimized with bioreactor cultivation so that the DHA content doubled up to 66.72 ± 0.31% w/w total lipids (10.15 g/L of DHA concentration) [18]. Maximum DHA yield (Y_p/x_) of 21.0% and 18.9% and productivity of 27.6 mg/L-h and 31.9 mg/L-h were obtained, respectively, in a 5 L bioreactor fermentation operated with optimal conditions and dual oxygen control strategy in *Schizochytrium* sp. [19]. Nevertheless, it is difficult for the wild-type (WT) strain to meet the requirements of industrial production due to the low biomass and *n*-3 LC-PUFA content, which accounts for the high cost of the downstream process [20]. Artificial mutagenesis has been applied to obtain high-yield DHA-producing strains for industrial fermentation. Ultraviolet (UV) radiation, a kind of non-ionizing radiation, causes gene mutation via maximum absorption by purines and pyrimidines present in DNA [21]. With UV irradiation, DHA percentage of the total fatty acids up to 43.65% was achieved using the mutant *Schizochytrium* sp. [22]. Therefore, UV radiation was used as a method for mutagenesis to obtain a *Schizochytrium* strain with a high yield of DHA. There is abundant research on the effects of salinity, pH, temperature, and media optimization on the DHA production. Nevertheless, the genome and transcriptome research of *Thraustochytrid* is still rarely reported. Transcriptome sequencing and comparative analysis of *Schizochytrium* mangrovei PQ6 at different cultivation times were presented by Hoang et al. [23]. Transcriptome analysis reveals that the up-regulation of the fatty acid synthase gene promotes the accumulation of DHA in *Schizochytrium* sp. S056 when glycerol is used [24]. Transcriptome and gene expression analysis of DHA producer *Aurantiochytrium* under low-temperature conditions were conducted by Ma et al. [25]. Zhu et al. Revealed the genome information of *Thraustochytrim* sp. [26].

De novo assembly of RNA-seq data serves as an important tool for studying the transcriptomes of “non-model” organisms without existing genome sequences [27]. Recently, transcriptome analysis has emerged as an essential method for the identification of genes involved in the secondary metabolites biosynthesis [28], such as the accumulation of fatty acids in the microalgae *Nannochloropsis* sp. [29], *Schizochytrium mangrovei* PQ6 [30], *Neochloris oleoabundans* [31], *Euglena gracilis* [32], and *Rhodomonas* sp. [33]. Recent research has indicated that DHA is synthesized by two distinct pathways in *Thraustochytrids*: The polyketide synthase (PKS) pathway and the fatty acid synthase (FAS) pathway [34]. Fatty acids are synthesized through the PKS pathway via highly repetitive cycles of four reactions, including condensation by ketoacyl synthase (KS), ketoreduction (KR), dehydration, and enoyl reduction (ER) [35]. Three large subunits of a type I PKS-like PUFA synthase in *Thraustochytrium* sp. 26185 have been identified [36]. According to the FAS pathway, small molecular carbon units can be polymerized to form chain fatty acids by fatty acids desaturases and elongases [37]. There are two families of desaturases, which are fatty acid desaturases (FADs) and stearoyl-coA desaturases (SCDs). Genomic and transcriptomic analysis revealed that both the FAS and PKS pathways of PUFA production were incomplete in *Thraustochytrids* strains [38]. The dehydratase and isomerase enzymes were not detected in the *Thraustochytrids* strain SZU445 [26]. Although FAD12, FAD4, and FAD5 have been reported in *Thraustochytrids*, some *Thraustochytrids* only contains the desaturase not belonging to the FAS pathway, such as FAD6 [39]. Previous research has illustrated that the DHA synthesis pathway in *Thraustochytrids* is different from the classic fatty acid metabolism pathway and remains ambiguous [40]. By comparing the transcriptome of wild type and the mutant, it could help us to elucidate the genes involved in the fatty acid enhancement and provide valuable information for clarifying the DHA synthesis pathway.

In this study, UV mutagenesis was utilized to obtain competitive *Aurantiochytrium* sp. strain with enhanced biomass and DHA production. The key genes related to the increasing DHA accumulation were explored by comparing the transcriptome between the mutant and the parent strain. 

## 2. Materials and Methods

### 2.1. Microbial Cultivation

*Aurantiochytrium* sp. PKU#Mn16 were previously isolated from mangrove (22°31′13.044″ N, 113°56′56.560″ E) from coastal waters in Southern China, and then maintained in the China General Microbiological Culture Collection Center (CGMCC). *Aurantiochytrium* sp. PKU#Mn16 was inoculated into M4 liquid medium made with 100% filtered natural seawater (from Mirs Bay in Shenzhen, China) containing glucose (2.00%), yeast extract (0.10%), peptone (0.15%), and KH_2_PO_4_ (0.025%) [41]. The seed inoculum of *Aurantiochytrium* sp. PKU#Mn16 was cultured in a shaking incubator (LYZ-123CD, Shanghai Longyue Equipment Co., Shanghai, China) at 23 °C and 180 rpm for 48 h. One hundred milliliters of medium in a 250 mL flask was inoculated with 5 mL (5% (*v*/*v*) inoculation ratio) of the above culture. Three biological replicates of each sample were examined.

### 2.2. UV-Mediated Mutagenesis

The microorganism solution was diluted 10^5^ times and applied to the plate. Then the microorganisms on the plate were mutagenized after 24 h of incubation in a constant-temperature incubator (LR-250, Shanghai Yiheng Technology Co., Ltd., Shanghai, China) at 23 °C. Before UV mutagenesis, the UV crosslinker (SZ03-2, Shanghai Netcom Business Development Co., Ltd., Shanghai, China) was turned on for 30 min to stabilize the light waves. The plates were placed in 0 W, 10 W, 20 W, 30 W, 40 W, 50 W, 60 W, 70 W, 80 W, and 90 W UV crosslinkers and irradiated for 0 s, 6 s, 9 s, 12 s, 15 s, 18 s, 21 s, 24 s, 27 s, 30 s, 33 s, and 36 s. After mutagenesis, the plates were incubated for 48 h in the dark, and then, the number of colonies was counted, and the lethality was calculated. Three biological replicates for each sample were examined. 

### 2.3. Biomass Determination

The mutagenized strain was cultivated as described in Section 2.1 for 48 h. The culture was then centrifuged (Z366K, HERMLE, Germany) at 10,000 rpm for 5 min to obtain the cell precipitate. After washing three times with deionized water, the cell precipitate was collected as the biomass and then lyophilized in a freeze dryer (Triad 2.51, Labconco, Kansas City, MO, USA) for 72 h. Three biological replicates for each sample were examined in the experiment.

### 2.4. Fatty Acid Extraction

Before the experiment, filter paper bags (Civil Administration Filter Paper Factory, Liaoning Province, China) were pretreated with a solvent mixture (chloroform:methanol ratio of 2:1 (*v*/*v*)) for 48 h and dried at 50 °C. Five hundred milligrams of freeze-dried cells were placed in a pretreated filter paper bag as a filter paper package and extracted in a Soxhlet extractor at 70 °C for 48 h (solvent as described above) [42]. Then, the filter paper package was dried and weighed. The difference between the weights before and after was the weight of the FAs. The remaining liquid was evaporated to dryness at 70 °C by a rotary evaporator. The FAs were rinsed completely with 5 mL of n-hexane and placed in a 10 mL glass tube [43]. Three biological replicates for each sample were examined.

### 2.5. Fatty Acid Structure and Composition Analysis

#### 2.5.1. Fourier Transform Infrared (FTIR) Spectrometer Analysis

KBr powder was uniformly mixed with the dried cells and compressed into a sheet (KBr to dried cell ratio of approximately 100:1). KBr was used as a background and detected using a Fourier transform infrared (FTIR) spectrometer (Thermo Fisher Scientific, Waltham, MA, USA). The infrared spectrometer had a spectral range of 7800–350 cm^−1^, and its scanning frequency was 65 spectra (16 cm^−1^ resolution). Three biological replicates of each sample were prepared [44].

#### 2.5.2. Gas Chromatography and Mass Spectrometry (GC/MS) Analysis 

The FAs obtained in Section 2.4 were added to 5 mL of a 4% sulfuric acid–methanol solution (*v*/*v*), and 100 μL of a nonanecene–methylene chloride solution (500 μg mL^−1^) was used as an internal standard. After the tube was allowed to stand in a 65 °C water bath for 1 h, 2 mL of n-hexane and 2 mL of deionized water were added, and the mixture was shaken for 30 s. Three biological replicates for the extraction were examined. Then, the upper organic layer was transferred to a new test tube, and the organic solvent was thoroughly dried with nitrogen. Finally, 1 mL of dichloromethane was added to each tube to dissolve the FAs, and the solution was then transferred to a chromatography bottle [45].

The FAs in the chromatography bottle were diluted 100-fold and analyzed by gas chromatography mass spectrometry (GC-MS, 7890-5975 Agilent, Santa Clara, CA, USA). The GC column for FA determination was HP-5MS (19091S-433) with a stationary phase of (5%)-diphenyl (95%)-dimethylpolysiloxane, constituting a weakly polar capillary column. The column had a maximum temperature of 350 °C and dimensions of 30.0 m × 250 μm × 0.25 μm. The GC inlet temperature was 250 °C, the carrier gas was high purity He, constant pressure mode was used, the head pressure was 1.2 psi, the split ratio was 10:1, and the injection volume was 1 μL. The column temperature rise program was determined by 37 FA mixing standards. The steps for selecting the peaks for the separation of the 37 FAs in the sequence were as follows. First, the temperature was raised to 180 °C at a rate of 25 °C min^−1^ from 60 °C, increased to 240 °C at a rate of 3 °C min^−1^, maintained for 1 min, and then heated to 250 °C at a rate of 5 °C min^−1^. The GC-MS transfer line temperature was 250 °C, and the mass spectrometer detector selected the full scan mode [38]. Three physical replicates for each sample were prepared.

### 2.6. Comparative Transcriptomic Analysis

#### 2.6.1. RNA Extraction and cDNA Library Construction

After the total RNA was extracted with TRIzol (Life Technologies, Thermo Fisher Scientific Inc.), an Illumina HiSeq 4000 system was used to construct the cDNA library. mRNA sequences were then selected and the library was prepared [46]. To assess the integrity of the total extracted RNA, an Agilent 2100 bioanalyzer was used. The preparation of two libraries of cDNA constructs and transcriptome sequencing was conducted by Huada Gene Technology Co., Ltd. (Shenzhen, China). Oligo (dt) magnetic beads were utilized for enrichment and purification of mRNAs from the total RNA of each sample. The purified mRNAs enriched were short fragments, which were reverse transcribed for first-strand synthesis, and the second strand was used for cDNA synthesis. Then, these obtained double-stranded fragments were ligated with adapters, and appropriate DNA fragments were used as PCR amplification templates.

#### 2.6.2. Illumina Sequencing, Assembly, and Annotation

cDNA library sequencing was carried out by an Illumina HiSeq^TM^ 4000, with 100 nt paired-end reads generated. The obtained reads were then filtered based on quality parameters of GC content, sequence duplication level, Q20, and Q30. High-quality clean reads were chosen from raw reads and reads with adapters and poly-N sequences were eliminated. De novo transcriptome assembly of clean reads was implemented through the Trinity assembly database program using default parameters [47]. Trinity software consists of three modules, namely, Chrysalis, Inchworm, and Butterfly (http://trinityrnaseq.sourceforge.net/). Initially, the Inchworm module formed a k-mer dictionary by breaking sequence reads (k-mer fixed-length sequence of k nucleotides, in repetition k = 25 bp). For contig assembly, the most recurrent k-mers were selected by removing low-complexity, error-containing, and singleton k-mers. Contigs were obtained until both side sequences could not protract with k-1 overlap. Then, the Chrysalis module was used to make the de Bruijn graph and gather the linear contigs. Finally, the Butterfly module was constructed to analyze de Bruijn graphs and produce transcript sequences. Transcript assembly was performed by using all generated contigs. The main transcripts that contained more than 200 bp were selected as uni-genes. BLASTX (Altschul et al., 1997) alignment was performed against public protein databases such as the non-redundant (Nr) protein (Deng et al., 2006), Kyoto Encyclopedia of Genes and Genomes (KEGG; Kanehisa et al., 2004), Clusters of Orthologous Groups (COG), Gene Ontology (GO), and Swiss-Prot (Ashburner et al., 2000) databases, and uni-sequences such as National Center for Biotechnology Information (NCBI) Taxonomy. KEGG is a database of metabolic pathways that is used to identify the gene products and functions associated with a cellular process. This pathway analysis provides a logical understanding of the complex biological performance of different genes in a network, and the analysis is performed by using BLAST software against the KEGG database. The cDNA sequence of mutant X2 was uploaded to GenBank (Accession number: MT232522).

#### 2.6.3. qRT-PCR Analysis

Total RNA was extracted using the TRIzol (Life Technologies, Thermo Fisher Scientific Inc.) method. For quantitative real-time PCR (qRT-PCR), primers were first designed according to transcriptomic sequence data using Primer Premier 5 software (Appendix A). Then, the SYBR Taq^TM^ Ex Premix (Tli RNaseH Plus) Kit (TaKaRa Japan) was used with the following thermocycler protocol: 95 °C for 30 s, followed by 40 cycles of 95 °C for 5 s and 60 °C for 30 s. The entire process was performed in a CFX96 BioRad RT-PCR detection system. Actin was used as a housekeeping gene, which helped us check for standard and normal gene expression. qRT-PCR was performed in 3 replicates, and relative gene expression was quantified using the 2^−ΔΔCt^ method [48].

#### 2.6.4. Statistical Analysis

Analysis of variance (ANOVA) was utilized for the statistical analysis of the data. The biomass yield and DHA productions of the wild type and mutant strains were analyzed by IMB SPSS Statistics 26.0 through a one-way ANOVA. The least significant difference (LSD) test was applied to determine the significant differences among the group means at *p* <0.05.

## 3. Results

### 3.1. Cell Mutagenesis 

To obtain a DHA-rich mutant with relatively high biomass yield, *Aurantiochytrium* sp. was subjected to random mutagenesis with UV irradiation. As shown in Figure 1, the fatality rate of the cells was sensitive to both UV treatment time and power. With a UV treatment time of 30 s, as the UV power increased from 10 W to 50 W, the survival rate decreased from 92.15% to 8.29%. The mutant treated with UV power of 50 W showed a rapid decrease in survival rate after 15 s (survival rate of 83.67%). The survival rate was 2.67% when the cells were exposed to UV (power of 50 W) for 33 s. The results showed that both the UV exposure time and UV power contributed to the severity of DNA damage in the cells. At present, UV radiation has been widely used in the breeding of microbial species [49], but rarely used in the production of DHA by *Aurantiochytrium* sp. Currently, researchers either chemically mutagenize strains or optimize fermentation conditions to increase DHA production. In 2014, Choi et al. optimized the extraction method of DHA to increase DHA production. In this study, acid catalyzed hot-water extraction of docosahexaenoic acid (DHA)-rich lipids from *Aurantiochytrium* sp. [50]. Cheng Yurong et al. (2016) performed mutagenesis of *Aurantiochytrium* sp. through cold stress (4 °C and FAS inhibitors (triclosan and isoniazid) to enhance DHA enrichment [51]. Shariffah et al. (2018) optimized the levels of fructose, monosodium glutamate, and sea salt through monosodium glutamate (MSG) experiments, predicting that DHA production by *Aurantiochytrium* sp. SW1 would reach 8.82 g/L [52]. Under the UV irradiation, the DHA content (0.20 g/g dry biomass) of *Schizochytrium* sp. increased by 38.88% compared with the parent strain [53]. Thus, based on the results, UV irradiation at 50 W for 30 s was chosen as the mutagenesis condition for breeding the DHA-producing mutant strain.

### 3.2. Screening of the Mutant Aurantiochytrium sp.

After UV irradiation, 135 colonies were obtained from the surviving cells. The first round of mutant screening was based on dry cell weight (DCW) enhancement. As shown in Figure 2a, 14 colonies (X1 to X14) exhibited significantly enhanced cell growth compared with the parent strain. Notably, the biomass yield of mutants X2 and X4 increased 1.53 ± 0.025 and 1.52 ± 0.053-fold, respectively. The lipid and DHA contents of the mutants were also analyzed (Figure 2a). Among the 14 mutants, eight independent colonies (X1, X2, X3, X4, X5, X9, X11, and X14) exhibited increased fatty acid yield (per g of DCW; 1.09 ± 0.056-fold, 1.79 ± 0.041-fold, 1.26 ± 0.043-fold, 1.25 ± 0.064-fold, 1.23 ± 0.024-fold, 1.35 ± 0.012-fold, 1.08 ± 0.033-fold, and 1.64 ± 0.059-fold, respectively). Four mutants (X2, X3, X5, X14) showed an improved ability of DHA production. In particular, mutant strain X2 showed a marked improvement of 1.90-fold compared with the WT strain. According to the cell dry mass, lipid, and DHA content, mutant strain X2 was chosen as the preferable DHA-producing candidate for the following experiments. 

To verify the hereditary stability of mutant X2, the strain was cultivated continuously in a shake flask for ten generations (Table 1). There was no significant difference for the DHA production observed among the ten generations. The DHA, lipid, and biomass yields of the tenth generation were 624.93 mg/L, 1417.37 mg/L, and 2920.60 mg/L, respectively. The results showed that UV irradiation (50 W, 30 s) could be utilized as a breeding strategy to screen for high-yield DHA-producing *Aurantiochytrium* sp.

### 3.3. PUFAs Production by the Mutant Aurantiochytrium sp.

Significant differences in FAs production were observed between the mutant and WT. As shown in Figure 2b, the amounts of LC-PUFAs (DHA (22:6, *n*-3) and EPA (20:5, *n*-3)) and saturated fatty acids (SFAs; hexadecanoic acid (HDA, 16:0) and pentadecanoic acid (PDA, 15:0)) were markedly different after UV mutation. The HDA and PDA levels decreased from 36.28% to 30.21% and 6.18% to 2.44%, respectively, whereas the DHA levels increased from 40.55% to 50.19%. The production of DHA in the mutant strain X2 increased by 23.77% compared with the WT. Interestingly, 9.07% of DPA was observed instead of EPA in the mutant X2. The results indicate that the mutation led to the transformation of SFAs to PUFAs, reflecting the mutated genes responsible for FAs carbon chain lengthening and unsaturation. Previous studies have confirmed that the accumulation of LC-PUFAs can be improved by increasing the SFA levels in the substrate, and that the long-chain saturated FAs of either C16 or C18 [54,55] could be further transferred to LC-PUFA by desaturase and elongase [56]. The culture conditions of mutant X2 were also studied to explore the appropriate conditions for DHA production. Figure 3 shows that pH 6.5, a fermentation volume of 200 mL, a culture temperature of 27 °C, and an inoculum size of 5% were suitable conditions for DHA accumulation in mutant X2.

### 3.4. Sequence Analysis and Assembly

For a comprehensive understanding of the molecular mechanism underlying FA improvement in the collection, a comparative transcriptomic study was conducted between the WT and the mutant X2. The Q20 base value with a base quality greater than 20 and an error rate ≤0.01% made up more than 96.82% and 96.67% of the WT and X2 reads, respectively, indicating that the raw sequence reads were very reliable and of high quality (Table 2).

After trimming the adapter sequences, ambiguous nucleotides, and low-quality sequences, the qualified mRNA-based sequenced reads were subjected to transcriptome de novo assembly (Table 3). For wild type Mn 16 and mutant X2, the transcriptome assembly generated 20,874 and 18,952 uni-genes with a N50 of 1880 and 2032 bp, respectively. The analysis of N50 indicated that 50% of the assembled reads were incorporated into transcripts more than 1880 and 2032 bp. The mean length of the transcripts was 1149 and 1205 bp.

### 3.5. Differentially Expressed Gene Analysis by RNA-Seq

In the comparison between WT and mutant X2, a total of 39,826 differentially expressed genes (DEGs) existed. Of these total DEGs, 1350 were downregulated and 1945 were upregulated in *Aurantiochytrium* sp. WT and mutant X2, respectively. Further elucidation of DEGs with different expression arrays was performed with hierarchical DEG clustering through Euclidean distance associated with complete linkage (Figure 4).

GO analysis of the transcriptome was based on three main categories: Biological processes, cellular components, and molecular functions (Figure 5a). In many cases, several GO terms were assigned to the same uni-gene. Biological processes, molecular functions, and cellular components related to functional subgroups were used to categorize all DEGs. DEGs in both WT and mutant X2 were implicated in cellular and metabolic processes, which are plentiful in biological processes. One of the proteins involved in catalytic and binding processes was the foremost protein in molecular function, and other cellular components included cell and cellular parts. 

DEG-associated pathways were analyzed by using KEGG pathway tools in *Aurantiochytrium* sp., with *p* <0.05 as the significance threshold. Nineteen significantly enriched pathways were found in WT and mutant X2 (Figure 5b). These enriched pathways were associated with lipid metabolism, carbohydrate metabolism, biosynthesis and translation, and secondary metabolite biosynthesis. KEGG pathways involved in lipid metabolism include fatty acid metabolism, glycerophospholipid metabolism, glycerolipid metabolism, fatty acid biosynthesis and secondary metabolite biosynthesis, which probably play an important role in PUFA biosynthesis.

### 3.6. Identification and Characterization of Genes Involved in DHA Biosynthesis

The expressions of the key genes in the PKS pathway have been determined for mutant X2, including CoA-transferase (*CoAT*), acyltransferase (*AT*), enoyl reductase (*ER*), dehydratase (*DH*), and methyltransferase (*MT*), shown as Table 4. Using transcriptomic sequencing, we identified only one fatty acid synthesis (FAS) desaturase encoded by a uni-gene. In addition, RNA sequence analysis was used to investigate key biosynthetic enzymes of PKS pathway genes (Table 4). This analysis reported that Unigene4591_All, Unigene2419_All, Unigene10491_All, CL663.Contig2_All, and CL555.Contig2_All were involved in synthesizing DHA, which was downregulated in Mn16 and upregulated in mutant X2.

### 3.7. mRNA Expression Level of the Mutant X2 and WT

qRT-PCR was used to check the DEG expression profiles associated with PKS pathways. mRNA expression levels were checked for different genes, such as *CoAT, ER, DH, MT*, and *AT*, in comparison with the levels in Mn16, showing downregulation in Mn16 but upregulation in X2 samples. The qRT-PCR results were consistent with the RNA-seq results (Figure 6) and validated the DEG expression profile.

## 4. Discussion

### 4.1. DHA Synthesis Enhancement

Recently, the DHA produced by microorganisms in the ocean has received increased attention [57]. Dietary supplements constitute the largest market share of 55% for *n*-3 products, followed by functional food and beverages, and pharmaceuticals [58]. The *n*-3-PUFA market is projected to show an annual growth rate of 12.8% between 2014 and 2019 and is expected to be worth USD 4300 million by 2019 (www.marketsandmarkets.com). DHA-rich oils from *thraustochytrids* are currently on the market as dietary supplements [59]. The main source of the marine *n*-3 fatty acids EPA and DHA are fish oils [60]. Approximately 200,000 tons of fish oils are used in products for the human markets. Meanwhile, the production of microbial *n*-3-rich oils constituted only 5000 tons in 2011, with marine protist *Thraustochytrids* and the heterotrophic microalgae *Chrypthecodinium cohnii* as production organisms [34]. Although industrial DHA production has been accomplished, certain approaches have been applied to enhance the synthesis of DHA via intrinsic [61,62] or extrinsic parameters [63]. Conversely, several shared features, including low adaptability, degeneration, and low production, continue to hinder significant production by strains [64]. An effective approach such as mutagenesis is broadly useful for selecting high-yield strains [65]. Alonso reported that microalgae produced high yields of DHA and EPA after mutagenesis, and increased EPA content was also observed in *Phaeodactylum tricornutum* mutated by UV light [66]. It was also reported that treating marine microalgae *Nannochloropsis salina* with UV mutagenesis, lipid accumulation of the mutant cultures was elevated to more than 3-fold that of the wild type strain. However, reduced growth rates resulted in a reduction in overall productivity [67]. Forjan et al. (2014) showed that UV-A increased the saturated:unsaturated fatty acids ratio, and hence an increase in storage lipids in *Nannochloropsis. gaditana* [68]. Liu et al. (2015) applied UV irradiation to microalgae *Chlorella sp.* and found that the biomass for the UV mutation strain was 7.6% higher and the lipid content reached the maximum value of 28.1% on day 15. Our research showed that in the mutant X2, the TFAs and DHA contents reached 81.73% and 35.24% of DCW, respectively. Results also prove that UV irradiation can be used as a breeding strategy for obtaining potential DHA-producing *Thraustochytrids* strain.

### 4.2. PKS Pathway

The DHA biosynthesis pathway in *Thraustochytriu* has not been fully elucidated. It has been reported that the two pathways, i.e., The PKS system and FAS pathway, they are likely to be present [69,70]. In general, eukaryotes biosynthesize the polyunsaturated fatty acids through a series of desaturation and elongation reactions catalyzed by membrane-bound enzymes such as desaturases and elongases, known as the fatty acid synthetase (FAS) system. A small amount of label in 22:6 was detected when the alga was grown in the presence of ^14^C labeled 18:0 or 18:1 [71]. The addition of ^13^C acetate or ^13^C butyrate in the growth medium resulted in 22:6, with only the odd carbon atoms enriched. The commonly found extended products of FAS in nearly all organisms are long-chain saturated FAs of either C16 or C18 [54,55]. The FAS pathway comprised seven or more kinds of desaturases, including ∆12, ∆9, ∆8, ∆6, ∆5, ∆4, and n-3 (e.g., ∆17 and ∆15). However, the activity of desaturase and elongase was not detected in *Schizochytrium* by ^14^C labeling, which implies the existence of a different DHA biosynthesis mechanism in the genus *Schizochytrium* [56]. The transcriptomic study of the mutant X2 and WT revealed that only one gene encoding desaturase was involved in the FAS pathway; however, the ∆4, ∆6, and ∆12 desaturase genes, which are important for DHA production, were not observed in the present study. Similar to previous studies, expressed sequence tag (EST) sequencing or PCR-based detection failed to identify the probable desaturases in the FAS pathway [72]. The modification of FAs is performed to produce long-chain DHA (C22:6) by an enzyme-dependent continuous process [73]. On the other side, it has been reported that marine bacteria could produce polyunsaturated fatty acid via the polyketide synthases (PKS) pathway [74]. PKS pathway domains are likely involved in the production of DHA, such as *AT, DH, MT* and *ER* [75,76]. Genomic and transcriptomic analysis have shown that *Thraustochytrids* contained some key enzymes of PKS system such as 3-ketoacyl-synthase (KS), ketoreductase (KR), and enoyl reductase (ER) [26]. As a probable source of high-value DHA, the PKS pathway is important in DHA biosynthesis, as genes related to the PKS pathway were mined in the transcriptome study of wild *Aurantiochytrium* sp. and mutant X2 (Table 4). These uni-genes were homologs to *MT, AT, ER* and *DH*, which are crucial in polyketide synthesis. These findings suggest that DHA synthesis is likely to occur via the PKS pathway in WT and mutant X2. Currently, no evidence supports the hypothesis that DHA biosynthesis occurs via either of the two hypothetical pathways [40]. Furthermore, the formation of PUFAs of >C22 (e.g., 28:8*n*-3 and 28:7*n*-6) also occurs via the PKS pathway, which has been described in some species of oceanic dinoflagellates by Mansour M.P. (1999) [77].

### 4.3. Transcriptional Responses of the PKS Pathway

The omega-3 PUFAs, including EPA, DPA, and DHA, are produced by certain strains, e.g., thraustochytrids [78]. Biochemical studies have been performed to characterize the distinct enzymes from the standard PKS pathways, which is ultimately helpful for understanding the underlying biosynthetic mechanisms [79]. These findings revealed that the FAS pathway does not participate in the biosynthesis of DHA in the *Aurantiochytrium* sp. strain. PUFA synthesis is carried out by ACP (acyl carrier protein) in the PKS pathway. ACP acts as a covalent joint for chain elongation during many cycles. The synthesis of lengthy (unsaturated) fatty acids includes several enzymes in the PKS system, e.g., *AT, MT, ER*, and *DH*. A vital role is played by *AT* domains and their allies, i.e., ACPs. *AT* loads the building units onto ACP (substrate acceptor). Therefore, *AT* decides which building blocks will be incorporated into the polyketide assembly [80]. *AT* plays significant roles in the PKS pathway; here, *AT* showed down-regulation in WT and up-regulation in X2 at the transcription level. The data showed increased gene expression of the PKS pathway in the mutant. PKS-linked genes contained *DH, AT, ER*, and *MT* domains, as revealed by the transcriptomic study of *Aurantiochytrium* sp. X2 (Table 4). The mutation led to the formation of ORFC (open reading frame control), which contains two *DH* domains and one *ER* domain, with up-regulation similar to that observed by Zhi-Qian Bi (2018) [81]. The mutation also improved the expression of *DH* and *ER* in *Thraustochytriu***,** which suggests increased production of DHA. It is believed that mutagenesis is a valuable strategy for the enhancement of PUFA biosynthesis. The PKS anchor gene up-regulation suggests that the PKS system is actively involved in PUFA biosynthesis, which is supported by [40].

## 5. Conclusions

UV mutagenesis enhanced the ability of DHA production and led to the generation of potential DHA-producing strain from *Aurantiochytrium*. The transcriptome of the WT and the mutant strain were compared to investigate the vital genes responsible for the DHA enrichment. Results showed that in both WT and mutant strain X2, FAS was incomplete and key desaturases, but genes related to the PKS pathway were observed. The qPCR revealed that the upregulation of key PKS pathway genes (*CoAT, DH, AT, ER, MT*) involved in the high yield of DHA for the mutant strain X2. The research provides valuable information for constructing a genetic engineering strain with rational design for the fatty acid composition in future work. 

## Figures and Tables

**Figure 1 microorganisms-08-00529-f001:**
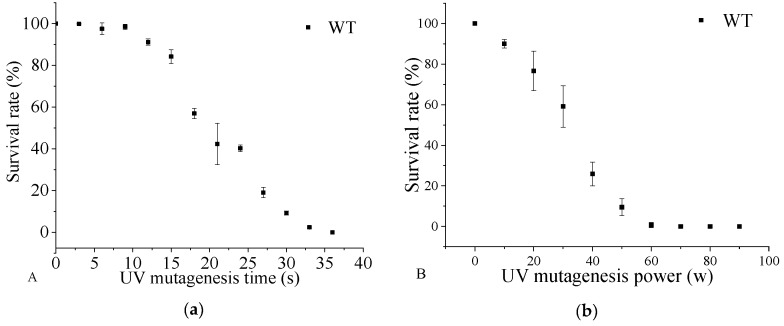
The survival rate of *Aurantiochytrium* sp. PKU#Mn 16 (wild type (WT)) under various ultraviolet (UV) mutagenesis time (**a**) and UV mutagenesis power (**b**). UV irradiation for 30 s (**a**) at 50 W (**b**) was selected as the WT mutation condition. All data were collected from three independent experiments. Error bars were the standard deviation.

**Figure 2 microorganisms-08-00529-f002:**
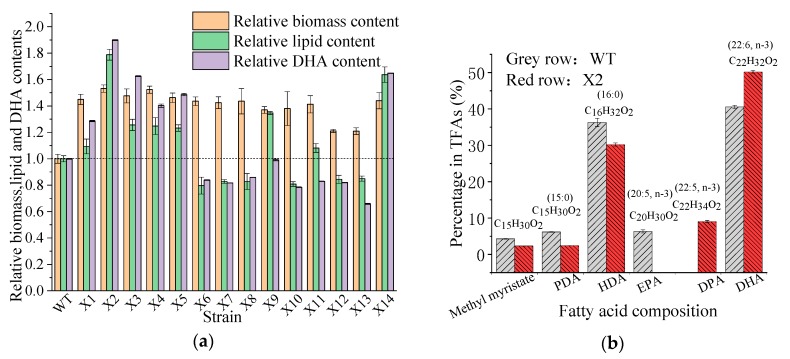
(**a**) The biomass, lipid and docosahexaenoic acid (DHA) contents of wild-type (WT) and 14 isolated mutant strains (X1~X14), and (**b**) the fatty acid component in total fatty acids (TFAs) of *Aurantiochytrium* sp. PKU#Mn16 (WT) and mutant strain *Aurantiochytrium* sp. X2 (X2). All data were collected from three independent experiments. Error bars represent the standard deviation. The values of biomass, lipid, and DHA contents of the wild-type strain were set to 1.0. PDA = Pentadecanoic acid, HAD = Hexadecanoic acid, EPA = Eicosapentaenoic acid, DPA = Docosapentaenoic acid, DHA = Docosahexaenoic acid.

**Figure 3 microorganisms-08-00529-f003:**
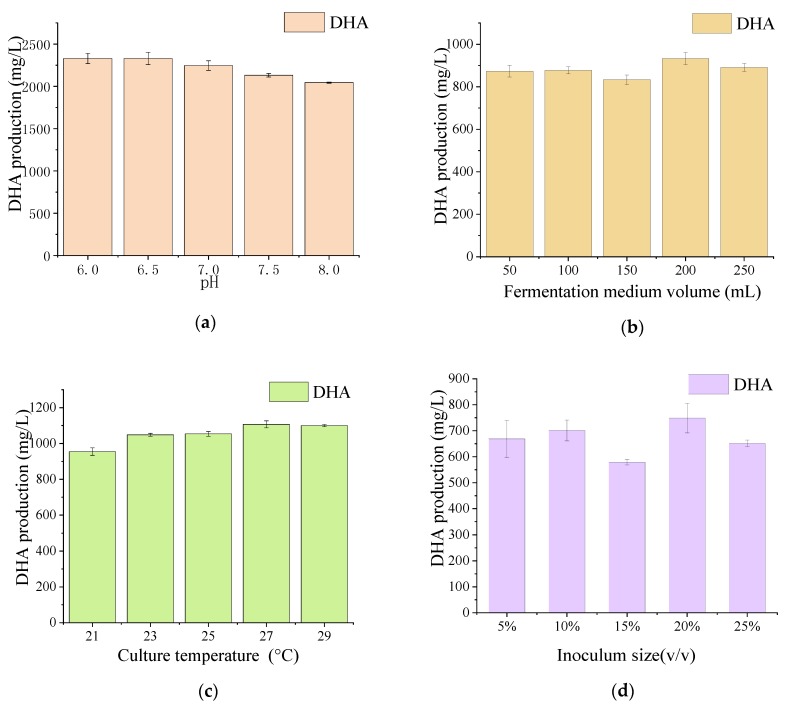
The effect of pH, cultivation temperature, fermentation medium volume, and inoculum size on the DHA production of mutant strain X2. The cultivation conditions were described as followed: (**a**) initial pH 6.0–8.0, culture temperature 23 °C, fermentation medium 100 mL, inoculum size 5%; (**b**) initial pH 6.5, culture temperature 23–29 °C, fermentation medium 100 mL, inoculum size 5%; (**c**) initial pH 6.5, culture temperature 23 °C, fermentation medium 50–250 mL, inoculum size 5%; (**d**) initial pH 6.5, culture temperature 23 °C, fermentation medium 100 mL, inoculum size 5%–25%. All data were collected from three independent experiments. Error bars were the standard deviation.

**Figure 4 microorganisms-08-00529-f004:**
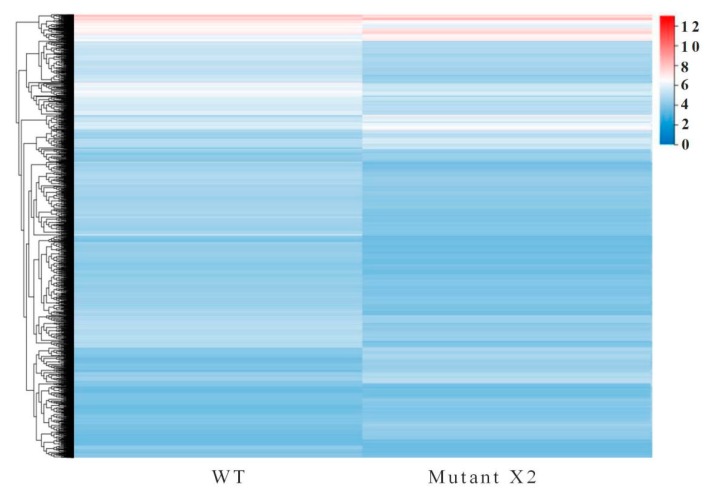
Heat map of the differentially expressed genes identified in *Aurantiochytrium* sp. PKU#Mn 16 (wild type) and *Aurantiochytrium* sp. X2 (mutant).

**Figure 5 microorganisms-08-00529-f005:**
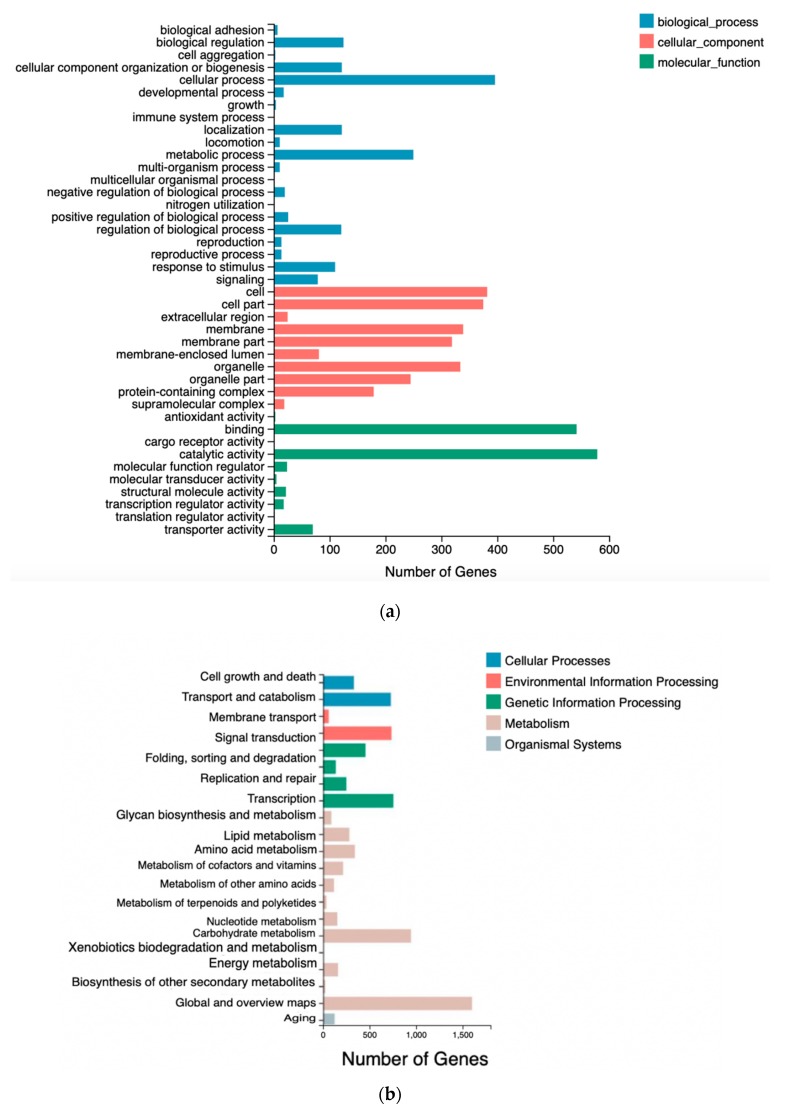
The differentially expressed genes (DEGs) of comparative transcriptomic analysis between wild type and mutant *Aurantiochytrium* sp.: (**a**) Ontology (GO); (**b**) Enriched Kyoto Encyclopedia of Genes and Genomes pathway (KEGG).

**Figure 6 microorganisms-08-00529-f006:**
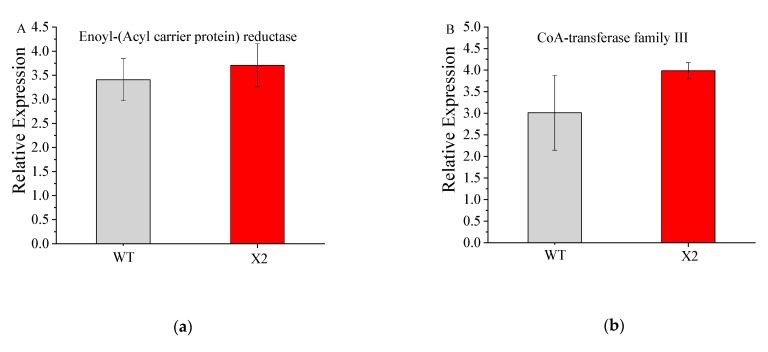
qPCR relative expression of genes annotated for enoyl-(acyl carrier protein) reductase (**a**), coA-transferase family Ⅲ (**b**), acyltransferase (**c**), methyltransferase (**d**), and dehydratase family (**e**) of wile type (WT; *Aurantiochytrium* sp. PKU#Mn16) and mutant strain (X2; *Aurantiochytrium* sp. X2). All data were collected from three independent experiments. Error bars were the standard deviation.

**Table 1 microorganisms-08-00529-t001:** The total fatty acids (TFAs) and docosahexaenoic acid (DHA) contents of mutant strain *Aurantiochytrium* sp. X2 during the ten-generations subculture.

Generation	TFAs (%DCW)	DHA (% TFAs)	DHA Content (mg/L)
WT	41.07 ± 2.05	40.55 ± 2.05	321.37 ± 14.98
X2-2nd	50.00 ± 3.87	44.39 ± 3.87	624.34 ± 2.95
X2-6th	48.10 ± 0.74	44.98 ± 0.74	623.40 ± 7.14
X2-10th	48.53 ± 1.40	44.09 ± 1.40	624.93 ± 13.32

The fermentation conditions during continuous subculture remained the same and were set as: Inoculum size 10% (*v*/*v*), culture temperature 23 °C, initial pH 6.5, and fermentation medium volume 1000 mL. DCW = dry cell weight, TFAs = total fatty acids, WT = wild type strain, X2-2nd = the second-generation subculture of mutant strain X2, X2-6th = the sixth-generation subculture of mutant strain X2, X2-10th = the tenth-generation subculture of mutant strain X2. All data were collected from three independent experiments.

**Table 2 microorganisms-08-00529-t002:** Summary of sequencing data for wild type *Aurantiochytrium* sp. PKU#Mn16 and mutant *Aurantiochytrium* sp. X2.

Sample	Total Raw	Total Clean		Clean Reads		
	Reads (Mb) ^1^	Reads (Mb) ^2^	Bases (Gb) ^3^	Q20 (%) ^4^	Q30 (%) ^5^	Ratio (%) ^6^
Mn16	50.94	44.69	6.70	96.82	88.53	87.72
X2	47.43	43.14	6.47	96.67	87.97	90.94

^1^ The reads amount before filtering; ^2^ The reads amount after filtering; ^3^ The total base amount after filtering; ^4^ The rate of bases in which quality was greater than 20 in clean reads; ^5^ The rate of bases in which quality was greater than 30 in clean reads; ^6^ The ratio of the amount of clean reads.

**Table 3 microorganisms-08-00529-t003:** Quality metrics of transcriptome and uni-genes assembly for wild type *Aurantiochytrium* sp. PKU#Mn16 and mutant *Aurantiochytrium* sp. X2.

Quality Metrics	Uni-genes	
	Mn16	X2
Total Number	20,874	18,952
Total Length	23,991,758	22,844,185
Mean Length	1149	1205
N50 ^1^	1880	2032
N70 ^2^	1264	1328
N90 ^3^	480	504
GC (%) ^4^	48.36	48.75

^1^ The N50 length is used to determine the assembly continuity. N50 is a weighted median statistic that 50% of the total length is contained in transcripts that are equal to or larger than this value. ^2^ N70 is a weighted median statistic that 70% of the total length is contained in transcripts that are equal to or larger than this value. ^3^ N90 is a weighted median statistic that 90% of the total length is contained in transcripts that are equal to or larger than this value. ^4^ GC (%): the percentage of G and C bases.

**Table 4 microorganisms-08-00529-t004:** The key candidate genes related to polyketide synthase (PKS) pathway wild type *Aurantiochytrium* sp. PKU#Mn16 and mutant *Aurantiochytrium* sp. X2.

Gene ID	Protein Name	Gene Name	Control FPKM	Treat FPKM	log2
CL555.Contig2_All	Enoyl-(Acyl carrier protein) reductase	*ER*	3.85	10.1	1.400242079
CL663.Contig2_All	CoA-transferase	*CoAT*	1.14	3.04	1.394908036
Unigene10491_All	Acyltransferase	*AT*	74.10	149.53	1.006525578
Unigene2419_All	Methyltransferase domain	*MT*	8.46	18.11	1.083865306
Unigene4591_All	Dehydratase family	*DH*	0.01	1.66	7.211944308
CL94.Contig2_All	Fatty acid desaturase	*FAD*	2.02	4.34	1.101442064

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
