# Peer review of "Comparative Transcriptomic Analysis Uncovers Genes Responsible for the DHA Enhancement in the Mutant Aurantiochytrium sp."

_microorganisms, 2020, doi:10.3390/microorganisms8040529_

Round 1
Reviewer 1 Report
please see attached file

Author Response
Thank you for your review, and the reply is attached.

Reviewer 2 Report
Line 28: add a sentence about the importance of DHA, before “Marine eukaryote……… Provide the abbreviations for all shortcuts for the first time you mention in abstract.
Line 31: what is X2??? Is it strain or isolate of Thraustochytriidae sp.?
Line 31: what is the yield of lipid and DHA without UV irradiation?? Add the values for comparison.
Line 34: delete “lacked”
Line 37: replace “potential genes ” with specific genes you observed.
Line 50: delete “floating”… add list of fish species if possible.
Line 53: delete comma after “content ,”
Line 56 -57: move this sentence to line 54, after reference 8.
Line 56:“doubled up to 66.72 ± 0.31% w/w total lipids ”… here add details on quantity per litre…
Line 58: delete “general”
Line 63: provide the mutant name??
Line 65: add reference for line 65.
Line 67: add previous reports.
Line 80: add reference
Line 87: add isolation source of Thraustochytriidae sp. PKU#Mn16
line 94-102: add reference for methodology you have followed..
Line 100: where is 30 s?? In abstract you have mentioned 50 W for 30 s. In methodology it is missing.
Line 104: replace “chapter” with “section”
Line 105 & 106: replace precipitate as “cell precipitate”
Line 110: delete “Hanjiang Road, Shuncheng District, Fushun City, Liaoning Province, China ”… add if company name is available…
Line 112: dried cells? Do you mean freeze dried cells. If so, please replace the sentence.
Line 109 to 117: add reference for methodology.
Line 126-133: add reference for this methodology.
Line 149-151: rephrase the sentence. Not in good order.
Line 158: DNA fragments were used as PCR amplification templates…. Provide these details in supporting file.
Line 166: delete fullstop after Butterfly.
Line 178: add the databases names.
Section 2.6.2: have you submitted the cDNA in genbank…. Provide the accession numbers. And mention these details in this section.
Section 2.6.4: provide the statistical details for DHA production. Which software used for statistics?? Detail.
Figures: all figure titles and additional details should be improved. Now figures are not so clear. Figure 2b is not clear. Instead of front and back… mention as black and red.
Line 234?: these values are not matching with table values.. cross check and confirm.
Line 253: Previous studies …. Add references.
Line 292: add respectively after.. X2 mutant Thraustochytriidae sp., respectively..
Figure 4: heat map is not clear… which one is WT and which one is X2… label the figure properly..
Line 303 to 307: place these lines in methodology.
Line 324: tABLE 1.. correct typographical error.
Line 390: remove underline.
Line 398: delete sp. and add strain name.
Line 403: add implications and future work on this mutant strain.
Line 404: I could not find supplementary files. Kindly provide the same.
Line 406 to 407: delete. Irrelevant.
Important suggestions: the work done in this paper provides new information. However, I strongly recommend authors to revise the manuscript with appropriate methodology they have followed. Most of the methodology mention here provide no references. Also, discussion part is not upto the satisfactory. Add the below given references. In discussion part, compare the DHA and other content with previous publications. May be a detailed table will be useful for quick grabbing of the comparison between your strain and previous studies. English correction needs to be done and shall be improved.
Add these references and improve the discussion and introduction parts:
https://www.ncbi.nlm.nih.gov/pmc/articles/PMC5995480/
https://www.researchgate.net/publication/258854232_Culturable_diversity_and_biochemical_features_of_thraustochytrids_from_coastal_waters_of_Southern_China
https://www.sciencedirect.com/science/article/pii/S0960852419316323
https://www.researchsquare.com/article/rs-14022/v1
Author Response
We would like to thank the Reviewer 2 for the careful and thorough reading of this manuscript and for the thoughtful comments and constructive suggestions, which help to improve the quality of this manuscript. the reply is attached.

Round 2
Reviewer 2 Report
authors have done good revision. i request authors to provide a supplementary table on DHA from Aurantiochytrium and its related family members.
Grammatical errors and rephrasing some of the sentences with appropriate words is suggested to carry out.
Author Response
We would like to thank the Reviewer 2 for the careful and thorough reading of this manuscript and for the thoughtful comments and constructive suggestions, which help to improve the quality of this manuscript. Our detailed response to comments follows.
